# Hole Depth Prediction in a Femtosecond Laser Drilling Process Using Deep Learning

**DOI:** 10.3390/mi14040743

**Published:** 2023-03-27

**Authors:** Dong-Wook Lim, Myeongjun Kim, Philgong Choi, Sung-June Yoon, Hyun-Taek Lee, Kyunghan Kim

**Affiliations:** 1Department of Mechanical Engineering, Inha University, Incheon 22212, Republic of Korea; 2Department of Mechanical Engineering, Chungnam University, Daejeon 34134, Republic of Korea; 3Department of Laser and Electron Beam Application Group, Korea Institute of Machinery & Materials, Daejeon 34103, Republic of Korea

**Keywords:** laser micromachining, deep learning, femtosecond laser, silicon nitride, blind hole

## Abstract

In high-aspect ratio laser drilling, many laser and optical parameters can be controlled, including the high-laser beam fluence and number of drilling process cycles. Measurement of the drilled hole depth is occasionally difficult or time consuming, especially during machining processes. This study aimed to estimate the drilled hole depth in high-aspect ratio laser drilling by using captured two-dimensional (2D) hole images. The measuring conditions included light brightness, light exposure time, and gamma value. In this study, a method for predicting the depth of a machined hole by using a deep learning methodology was devised. Adjusting the laser power and the number of processing cycles for blind hole generation and image analysis yielded optimal conditions. Furthermore, to forecast the form of the machined hole, we identified the best circumstances based on changes in the exposure duration and gamma value of the microscope, which is a 2D image measurement instrument. After extracting the data frame by detecting the contrast data of the hole by using an interferometer, the hole depth was predicted using a deep neural network with a precision of within 5 μm for a hole within 100 μm.

## 1. Introduction

A semiconductor test probe is a component that connects a semiconductor chip and test equipment to check the operation of the semiconductor die. When it makes contact with the pad or solder bump on the wafer, the probe pin located on the probe card, which is a consumable component, delivers electricity and evaluates if the chip is good or defective according to the signal returned at the moment. The substrate is drilled to provide support for the probe tip. Recently, as the degree of substrate integration has been enhanced, the spacing between holes has shrunk to 40 μm or less, the hole diameter has shrunk to 30 μm, and approximately 20,000 holes are now required per board.

Compared to mechanical hole processing, hole processing using a laser has the advantage of a shorter manufacturing process and easy re-processing when the probe card is worn out. Furthermore, a laser enables micro-hole processing and is simple to process with a high degree of substrate integration. [1,2]. The work that requires high integration of laser processing is sensitive to changes in mechanical factors such as deformation of the processing board owing to heat. Because heat does not affect the substrate during femtosecond laser processing, a femtosecond laser is essential for heat-sensitive microfabrication. [2,3]. Also, since it has a short pulse duration time of 10^−15^ s and generates high peak power with low pulse energy, it is widely used in laser micro-processing. In addition, recently, studies on improving the fatigue and corrosion performance of materials by induction surface modification technology using femtosecond lasers [4,5] and laser cleaning studies on the breakdown of air with high peak energy have been conducted [6].

Because the result value according to the variable seems nonlinear, it is difficult to anticipate the outcome under any processing conditions because of the nature of laser processing [7,8]. Furthermore, laser processing may produce varying results owing to external environmental factors. To manufacture high-quality probe cards, machining defects caused by numerous factors must be compensated by employing accurate values of the process variables.

The quantity of light incident in a hole diminishes as the depth of the hole increases, making the depth measuring of a blind hole in laser micromachining difficult with an eye-dependent measurement method such as an optical microscope. There is also a way of directly examining the cross-section of the blind hole by polishing [9], although objective depth measurement is problematic owing to grinding parameters. To objectively and accurately measure, numerous measuring tools and methodologies have been proposed [10,11,12,13].

Shetty [10] demonstrated a novel optical methodology for detecting the existence and depth of blind holes in turbine blades. The depth and width of the holes were measured based on the angle and time of the reflected beam after an obliquely incident laser beam was irradiated into them. A new optical measurement approach for an aircraft jet engine’s blind hole was presented, and because it is a way of measuring one hole in one process, productivity is low for procedures that need approximately 20,000 holes to be measured.

Wu [11] investigated the depth measuring of through holes in the field of automatic drilling and rivets. A probe was placed in a hole, and the depth was measured by receiving the laser signal according to the location via the charge-coupled device (CCD) sensor. The depth parameter may be correctly measured using the approach presented in the study, and the accuracy and stability required to fulfill the depth measuring criteria for automated drilling and riveting of large composite board pieces were proven. However, the probe diameter was 2 mm, making it unsuitable for detecting the depth of micro-holes.

White light interferometry is a high-precision measuring technique that produces three-dimensional geometric data such as length, surface profile, roughness, and depth. [12]. Mezzapesa [13] used a laser self-mixing interferometry approach to address the drawbacks of the existing double-arm interferometer method to measure ultra-high-speed laser drilling holes in a metal plate with a resolution of 0.41 μm. The hole depth measurement method using a white light interferometer had a very fast response time and high resolution measurement accuracy, but the system configuration, such as the use of a variable wavelength laser, the configuration of an optical system for an interferometer, and high-precision measurement equipment, is complex and expensive.

Because of its low costs, high speed, automation, and vast area processing, machine vision is utilized in the field of hole inspection [14]. Wang [15] created an automatic optical inspection method based on machine vision to recognize the positions, defects, and absence of holes on printed circuit boards (PCBs). After labeling the image, basic dimensions such as hole area, circularity, center, opening, etc. were calculated and the basic dimensions and matching methods were used to identify the errors, ranges, omissions, etc. of the hole. It was discovered to have good performance in measuring two-dimensional geometric data such as hole diameter and position in the case of machine vision. It is not, however, appropriate for acquiring information in the depth direction.

Ho [16,17] used machine vision to detect glass surface profiles and defects during a laser-cutting process. The image pre-processing process and the use of deep learning techniques enhanced the detection process of defects in glass. It is significant that the glass defect was predicted using machine vision.

In this study, an interferometer was used to measure the hole processing depth while the laser processing parameters were varied; a deep learning model was devised for predicting measured data using 2D images. The light that enters the hole is continually reflected and absorbed by the inner wall of the hole; the deeper the hole, the less reflected would be the light. Therefore, the depth of the hole may be estimated using data from an optical microscope measuring the quantity of light entering and being reflected from the inside of the hole. Uniform lighting and an exterior environment are required in this regard. Furthermore, the hole diameter parameters were analyzed in this study, together with the amount of light, to improve the discriminating power of the model.

## 2. Experiment

### 2.1. Configuration of the Laser Processing Equipment

In this experiment, silicon nitride (Si_3_N_4_) was used as the probe card material. The thickness of the silicon nitride used was 200 μm, and the laser focus was on the surface of the silicon nitride. To perform the machining process, a Yb:KGW laser (Pharos, Light Conversion, Vilnius, Lithuania) with a pulse width of 233 fs was used. Furthermore, a UV wavelength was used, and the pulse energy used was 2.6 μJ. The processing equipment configuration is shown in Figure 1a. It can be seen that the laser light passes through the optical system and the Galvano scanner (IntelliSCAN10, Scanlab, Munchen, Germany) to process the specimen. Figure 1b shows a schematic design of the processing equipment structure. A computer is used to control the laser and scanner. A laser repetition rate of 50 kHz and an output power of 130 mW were used to process a 50 µm diameter hole. The laser drilling method used a trepanning method, considering the hole diameter. Hole processing was carried out by varying the number of processing cycles from 1 to 20, in order to predict the depth of the processed hole based on the number of processing cycles.

### 2.2. Effective Diameter and Depth Measurement Method for Hole Processing Geometry Analysis

Accurate measurement of actual measurement data is necessary to forecast the depth of 2D images by using deep learning algorithms. The shape of the hole produced in silicon nitride was measured using an optical microscope (Measuring microscope MM-800, Nikon, Tokyo, Japan) and an interferometer (WI-001, Keyence, Osaka, Japan). An optical microscope with coaxial illumination (epi-illuminator) was used as shown in Figure 2a. The diameter was measured as shown in Figure 2b. In addition, the optical microscope was utilized to determine the depth of the hole using the defocus measurement technique (DMM) [18], and in this method, when the depth of the hole exceeds roughly 150 μm, the bottom of the hole is not focused, making measuring impossible. An interferometer, as illustrated in Figure 2c, was utilized to quantify the objective hole depth based on the number of processing cycles; it is a laser-based method for determining depth, utilizing the interference effect. The profile of the hole processed by the interferometer is shown in Figure 2d. This hole-processing shape data was digitized to determine the hole depth. Interferometers have a fine depth data resolution of less than 1 μm; however, because the diameter resolution is less than 4 μm, accurately measuring the hole diameter is difficult. The depth of the processed hole was determined using the minimum value among the depth profile data to check the similarity between the depth obtained by interferometry and the DMM approach using an optical microscope. The measurement technique generated the most comparable value to the DMM approach by utilizing the minimum profile value in the graphs in Figure 2e,f.

### 2.3. Processing Hole Shape According to Laser Processing Frequency and Power

Experiments were carried out by varying the laser strength to generate blind holes in silicon nitride. When the laser intensity is increased, a hole deeper than the thickness of silicon nitride is formed, resulting in a through hole rather than a blind hole. Furthermore, if the laser power is low or the number of processing cycles increases, the peak energy needed to generate ablation in silicon nitride is insufficient, and melting occurs inside the hole owing to heat accumulation as the number of processing cycles increases. Figure 3a shows the rear side of the silicon nitride as the laser power was increased from 130 mW to 155 mW at pulse energy 2.6 μJ to 3.1 μJ. The peak power of the laser pulse was about 1.1 × 10^7^ W to 1.3 × 10^7^ W. When the power of the laser was 145 mW and 155 mW, the rear side of the silicon nitride was pierced; however, when the power was set to 130 mW, the rear side of the silicon nitride was not pierced. Figure 3b shows the results of processing from 1 to 100 under a power of 130 mW. In Figure 3b, molten material was visible within the holes processed 25 to 100 times, and analysis of the molten material by SEM-EDS revealed that the material and composition were consistent with silicon nitride. In Figure 4a, the depth was measured according to the number of processing cycles when the laser power was 82 mW, 105 mW, and 130 mW. It was proven that the depth was not constant owing to melting inside the hole, as illustrated in Figure 4b, when the number of processing cycles was 15 or more at 82 mW and 105 mW. Therefore, the hole was processed under a laser power of 130 mW.

### 2.4. Image Analysis According to Brightness, Exposure Time, and Gamma Value

To forecast the depth of a hole under multiple conditions using 2D images obtained using a microscope, a distinction between the features of each condition and the surrounding conditions is required. Therefore, the dynamic range of the contrast within the hole according to the brightness, which is the lighting condition, was analyzed to optimize the gray scale data, which are the contrast data of the image. The dynamic range was determined as the gray scale difference between the depths of the shallowest hole and the deepest hole; the deepest hole had been processed 20 times. Figure 5a shows the dynamic range when the lighting brightness was adjusted from 5 to 10. The measurement revealed that the brighter the illumination, the higher the dynamic range. The amount of light reflected by incident light into the hole reduces as the depth of the hole increases. At this point, increasing the quantity of light incident into the hole by altering the brightness of the light increases the amount of light detected. Figure 5b shows the depth contrast data for each number of laser treatments at the highest lighting brightness level of 10. Because the amount of light reflected from the inside of the hole is restricted, the contrast data converges after around 10 repetitions of laser processing.

The dynamic range of the contrast inside the hole was investigated in relation to exposure duration and gamma correction value. As the exposure period grows, the amount of light that had been gathered and the brightness of the image also increase. Figure 6a compares the measurement data left and right images of the depth of the shallowest hole and the depth of the deepest hole, which was processed 20 times, under 10, 30, 50, 70, and 80 ms exposure time conditions. Background areas are dominant at 10 ms and remained at the borders even at 30 ms. Only the contrast inside the hole remained in the image after 50 ms, and data began to be lost when the depth was shallow after 70 ms. As depicted in Figure 7a, the image results were normalized with the highest gray value according to exposure time and shown in arbitrary units (AUs) to compare with the depth of the actual machined hole. The condition with the highest similarity to the actual hole measurement value was 50 ms, and the value was almost linearly decreased according to the number of laser repetitions. When the slope of each data set was compared to the actual hole depth, the condition of 50 ms was determined to be the closest. The dynamic range expands when the brightness of the image increases but decreases at on exposure time of 50 ms. Furthermore, it was discovered that under the condition of 80 ms, a significant amount of light from the processed section was collected, making it difficult to view the processed shape.

Gamma correction employs a nonlinear transfer function to modify the intensity of light, and image discernment can be improved by selecting an appropriate gamma value. Figure 6b shows a comparison of the measurement data left and right images of the depth of the shallowest hole and the depth of the deepest hole, which was processed 20 times under each gamma correction condition based on a 50 ms exposure duration. Figure 7b shows the outcome based on the gamma value. It demonstrated the most comparable tendency to the actual hole measurement value under a gamma of 140, and the slope value was also the closest to the actual measurement. Therefore, a brightness of 10, exposure time of 50 ms, and gamma of 140 were selected as optimal conditions to maximize hole depth discrimination.

### 2.5. Data Preprocessing

A masking technique was performed to eliminate the data from the unprocessed region in order to quantitatively discern solely the contrast inside the hole. An area of 170 × 170 pixels was retrieved by cutting away the outside region, using a square area in the middle of the hole as a mask and averaging the contrast data. The processed data and label data were arranged into a 10 × 20 × 3 data frame based on the experimental sequence and circumstances. When the data set of contrast characteristics is shown as a scatter plot, as illustrated in Figure 8a, data that are clearly distinguished from other points are included. These values should be considered outliers and eliminated. In two steps, outliers were discovered using the standard deviation approach, and then values with a standard deviation of ±2 or more were selected to remove outliers near the mean. Missing value processing was then performed for the removed outliers. Missing values were replaced with the average value of the same processing condition data. Figure 8b shows the data after the outliers have been removed and missing values have been processed. In this method, the imported data were scaled to reduce the scale gap between the image processing input data and the carefully measured label data. As a scaling approach, the Z-score normalization was employed to normalize all characteristics so that they all had a distribution with a mean of 0 and a standard deviation of 1.

## 3. Deep Learning Study

### 3.1. Model Construction

To develop a prediction model, a deep learning model was built and trained in Python using TensorFlow and Keras. The Keras model employs a sequential model in which layers are successively built, with a Dense layer serving as the layer. Each neuron in the Dense layer gets inputs from all neurons in the preceding layer and adds weights that link the inputs and outputs. The activation function was the hyperbolic tangent function, a nonlinear function with two input characteristics, one output characteristic, and two hidden layers.

In the compilation step of setting up the learning process of the model, the optimizer, loss function, and metric used were the adaptive moment estimation (Adam), mean squared error (MSE), and mean absolute error (MAE), respectively.

The optimizer is responsible for determining the best possible outcome at the lowest possible cost, based on the actual result and the result predicted by the model during the learning phase using the training dataset. The Adam optimizer was developed by combining the benefits of RMSProp, which adjusts the learning rate, and Momentum, which changes the route, and is the most often used optimization technique. The loss function seeks to discover the weights that comprise the loss function throughout the training phase. The MSE is calculated by averaging the squared difference between the actual and anticipated results, and it has the benefit of highlighting the area where the error is most noticeable. The metric is an evaluation index that decides which index to examine while evaluating performance during the model verification step. Because the MSE computes the square of the error, it is sensitive to outliers because the real error becomes larger than the average error. The square root of the MSE was used to calculate the root mean square error (RMSE), which may be used to mitigate these drawbacks. The MAE is calculated by considering the difference between the actual and expected results as an absolute value and then averaging it. Of the 200 data points, 140 were classified as training data, and 60 data were classified as the test data. The built model was trained for 200 epochs using the training data, and the trained model was tested for loss, the MAE, and the RMSE using the test data to verify the model.

### 3.2. Hyperparameter Optimization

To improve the model’s performance through hyperparameter optimization, the MAE of the model built by modifying the number of layers and the number of nodes in each layer was compared. By adjusting the optimizer’s learning rate, optimal circumstances were discovered.

The model’s structure was determined through the addition of a hidden layer. The first and last layers were fixed at two and one nodes, respectively, to identify the structure of the model, and the experiment was conducted by varying the number of hidden layers and the number of nodes in each hidden layer. The experimental settings and results are given in Figure 9 and Table 1. The MAE for the training data reduced as the number of hidden layers and nodes grew, whereas the MAE for the validation data increased. It is determined that overfitting happened as the model’s complexity increased. The two layer–four node model with the lowest validation MAE value was used for continued learning.

To determine the appropriate learning rate for the optimizer, we performed two procedures, and it was confirmed that the MAE value for the test data was dropped at a learning rate of 10^−3^–10^−1^ when the model was generated by randomly specifying 100 integers in the range of 10^−6^–10^−1.^ The MAE value for the test data was acquired with a model developed by randomly defining 100 values in the range of 10^−3^–10^−1^ to obtain a more accurate learning rate. Finally, it was determined that the model performed best at learning rates ranging from 0.08 to 0.09; thus, the model’s learning rate was set to 0.08.

### 3.3. Learning Result

Learning was carried out across 200 epochs using preprocessed data and the model chosen for optimization. The MAE value according to the period is shown in Figure 10a, and a final score of 0.2172 was attained. Because this is a normalized result, the MAE value of the real data may be determined by multiplying the standard deviation of this data by 6.97. This indicates that the model trained on the hole image’s contrast and diameter data can estimate the real hole depth with an average error value of 6.97 μm. Figure 11a compares real data to data predicted by the learning model, with the dotted line indicating the ±5 μm point. As shown in the picture, the forecast error increases in the part where the actual machining depth is significant. This is apparently because, as the number of process cycles and hole depth increase, the contrast difference between adjacent situations reduces, and the discriminating power decreases. Looking at Figure 8b, while the data under the same processing settings had a contrast deviation of roughly five, data in a region with a high number of iterations are difficult to distinguish using the learning model because the contrast difference in the proximity condition is only one to two.

By modifying the input data, the number of processing cycles was 1 to 10, and the same model was trained. Figure 10b shows the MAE value according to the epoch. A final value of 0.1566 was obtained. This indicates that the actual hole depth can be predicted with an average error of 3.86 μm, and the prediction accuracy increases as compared to learning the current data 1 to 20 times. Figure 11b shows that pictures within the real depth of 100 μm are accurately predicted; however, at depths beyond that, the inaccuracy becomes predominant and the number of points out that beyond the green dotted line, which is the 5 μm increases. A depth of 100 μm corresponds to an aspect ratio of two based on a machined hole diameter of 50 μm, and it was demonstrated that a hole with a lesser depth may be predicted with less than 5 μm accuracy.

## 4. Conclusions

The silicon nitride support plate of a probe card was treated using a femtosecond laser for each number of processing cycle to obtain 2D images, and an experiment was carried out using deep learning to forecast the real hole depth based on data acquired by 2D photographs. The number of processing cycles with the gray scale value, which is the contrast data value of the 2D picture, was used to predict the depth of the hole.

The detection of incident light reduced as the depth of the hole increased, making it more difficult to examine the 2D picture. Therefore, an experiment was performed to gather a large amount of light by altering the brightness of the light to 5 to 10 and the exposure period to 10 to 80 ms. As a result, the light’s maximum brightness was 10 and the dynamic range displays the highest value for an exposure duration of 70 ms. Also, experiments were carried out by increasing the gamma value to 100, 140, and 180 to get a comparable gray scale data distribution with the data measuring the depth of the machined hole. The gamma value helps correct the picture by employing a nonlinear light intensity transfer function. When the exposure period is 50 ms and the condition is 140, the gamma value exhibits the most comparable data distribution to the actual hole depth measurement value.

Learning was conducted over 200 epochs using preprocessed data and a Python-based prediction model. For the holes that were machined 1 to 20 times, the depth was estimated with an average inaccuracy of 6.97 μm. The depth was estimated from 1 to 10 with an average inaccuracy of 3.86 μm. In addition, the model predicts holes with a depth of 100 μm or less with an accuracy of 5 μm or less. This study affords a method for predicting the hole processing depth using 2D pictures obtained by analyzing laser-machined holes with a microscope. Research is underway to realize universal prediction by securing data based on the varied dimensions of the machined holes.

## Figures and Tables

**Figure 1 micromachines-14-00743-f001:**
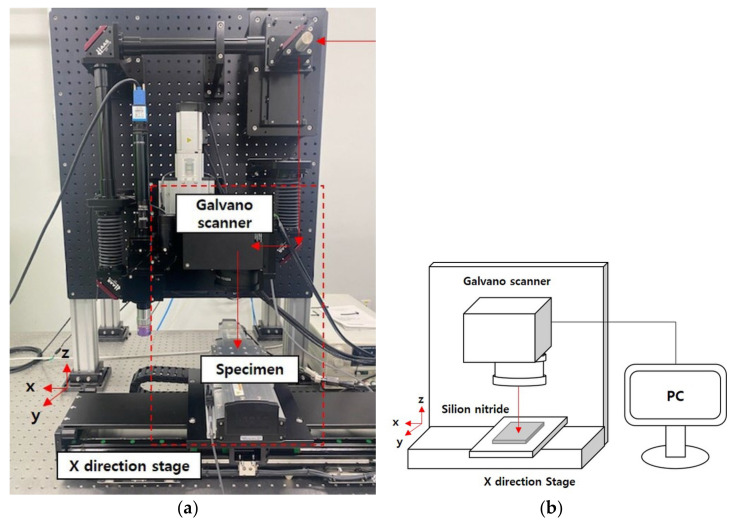
Equipment configuration for laser processing: (**a**) actual equipment for laser processing; (**b**) schematic of laser processing.

**Figure 2 micromachines-14-00743-f002:**
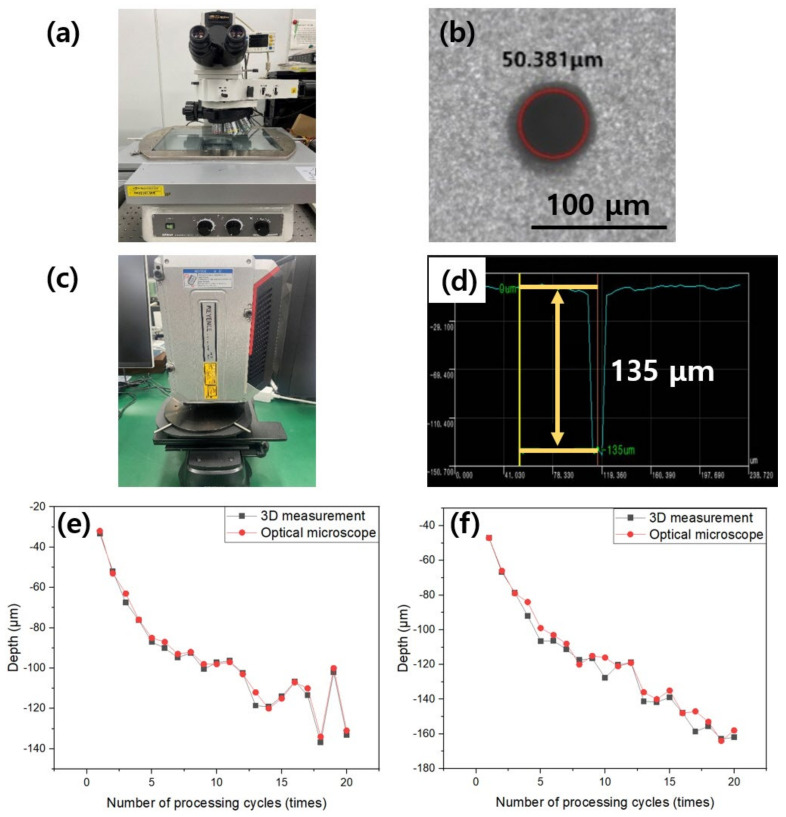
Equipment for measuring diameter and depth, as well as a method for analyzing hole processing geometry. (**a**) Optical microscope for diameter measurement; (**b**) diameter measurement image using an optical microscope; (**c**) interferometer for depth measurement; (**d**) depth profile measured using an interferometer; (**e**) comparison of optical microscope and interferometric depth measurement according to the number of processing cycles under a of laser power of 105 mW; (**f**) comparison of optical microscope and interferometric depth measurements according to the number of processing cycles under a laser power 130 mW.

**Figure 3 micromachines-14-00743-f003:**
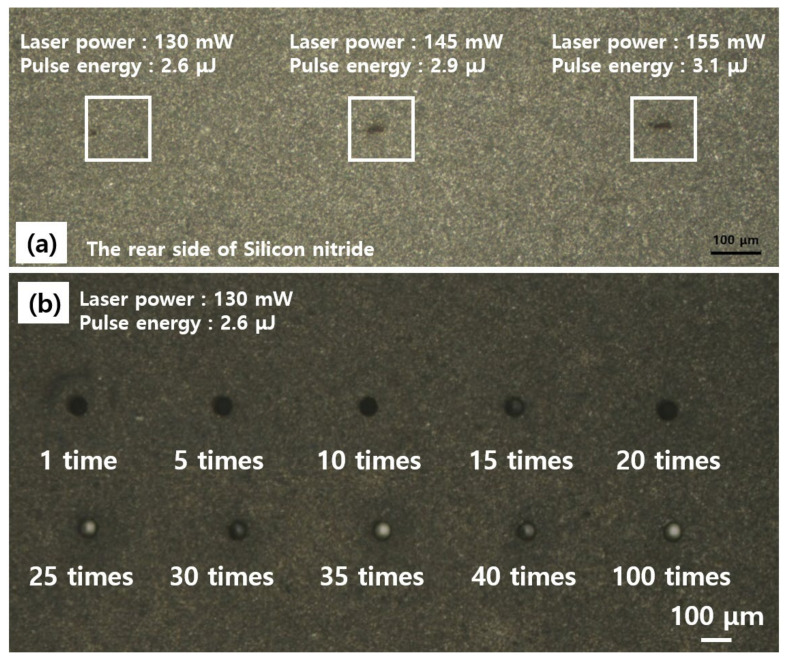
(**a**) Images of the back side of silicon nitride comparing the penetration power at laser power of 130 mW, 145 mW, and 155 mW; (**b**) comparison of holes obtained after 1 to 100 processing cycles at laser powers of 130 mW.

**Figure 4 micromachines-14-00743-f004:**
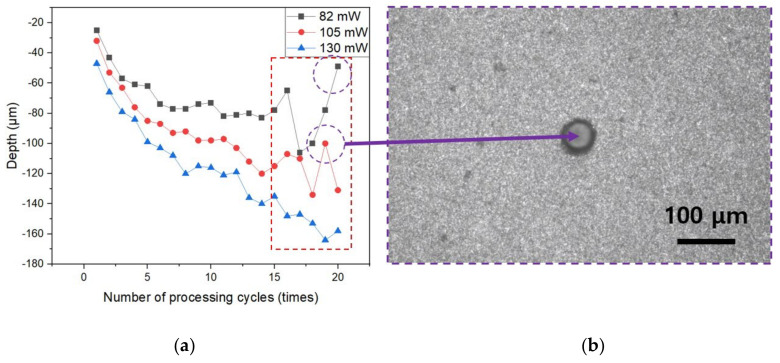
(**a**) Comparison of processing hole depth according to laser power and number of processing cycles; (**b**) observation image of molten material inside the processing hole for the outlier in (**a**).

**Figure 5 micromachines-14-00743-f005:**
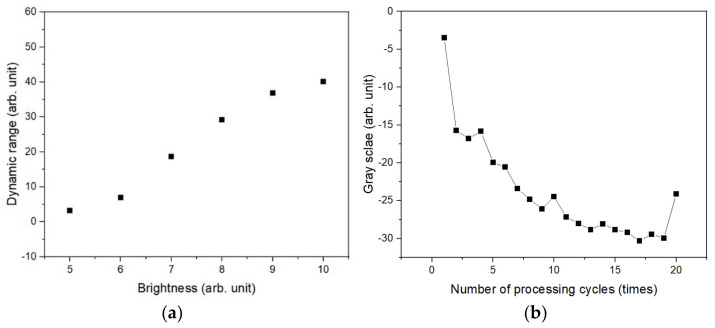
Dynamic range based on light intensity and number of laser treatments: (**a**) change in dynamic range as a function of light brightness; (**b**) gray scale and dynamic range as a function of the number of laser treatments and the hole processed under a laser power of 130 mW.

**Figure 6 micromachines-14-00743-f006:**
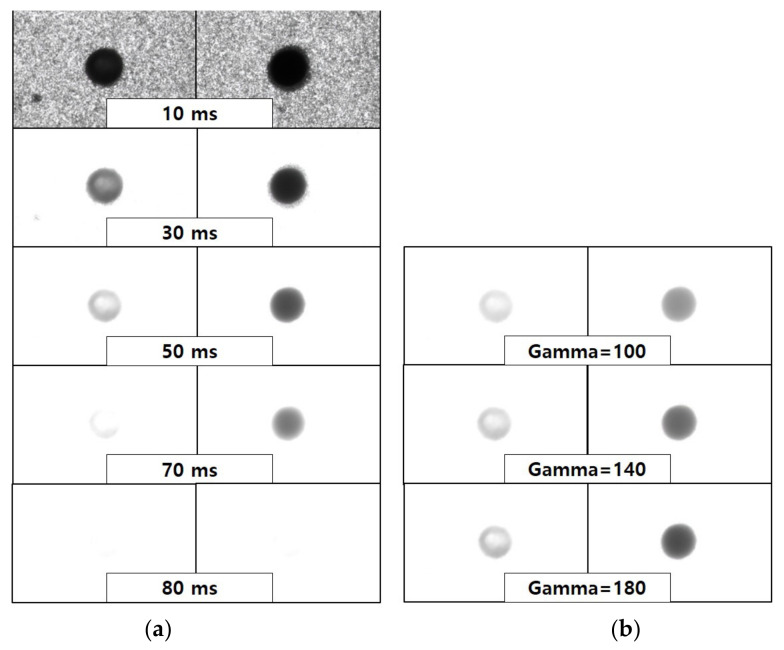
(**a**) Image photograph according to the exposure time of the optical microscope. (**b**) Image picture according to the gamma value.

**Figure 7 micromachines-14-00743-f007:**
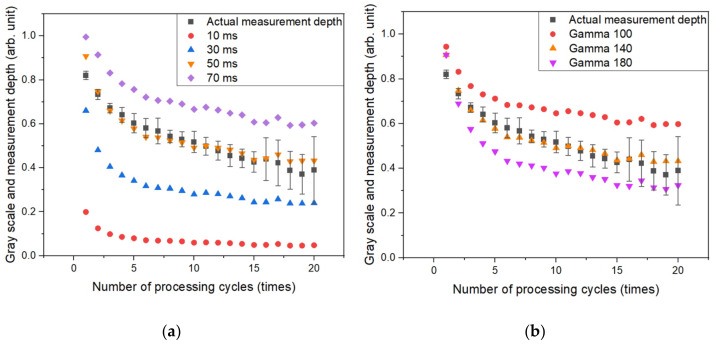
(**a**) Correlation between the real hole depth and gray scale according to optical microscope exposure duration. (**b**) Correlation between the real hole depth and gray scale based on the optical microscope gamma value.

**Figure 8 micromachines-14-00743-f008:**
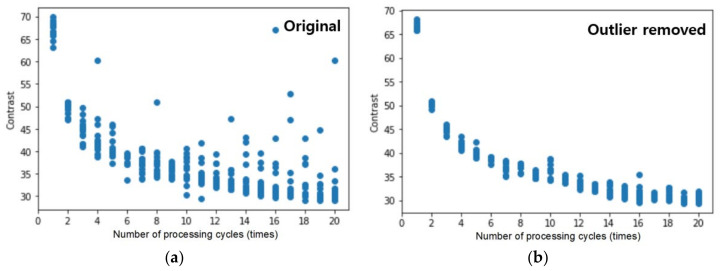
(**a**) Scatter plot of the intensity data by processing number, including outliers and missing values; (**b**) scatter plot of the intensity data by processing number, with outliers and missing values eliminated using the standard deviation approach.

**Figure 9 micromachines-14-00743-f009:**
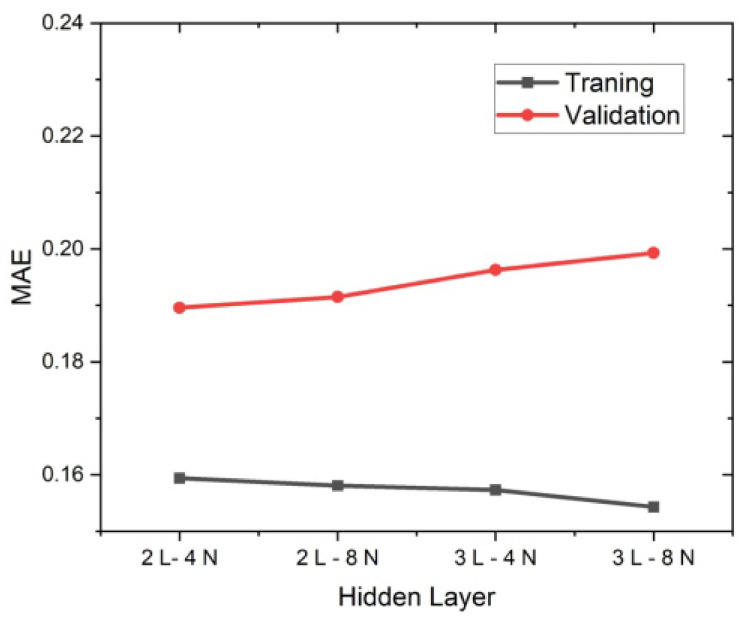
Results of training MSE and validation MAE according to the condition of the number of nodes in the hidden layer.

**Figure 10 micromachines-14-00743-f010:**
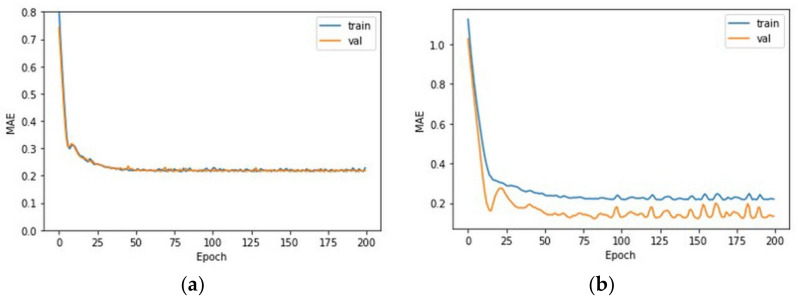
Outcome of selecting the best model and training it for 200 epochs. (**a**) An MAE result of 0.2172 was obtained based on the epoch for the condition of 1 to 20 processing times. (**b**) An MAE result of 0.1566 was obtained based on the epoch for the condition of 1 to 10 processing times.

**Figure 11 micromachines-14-00743-f011:**
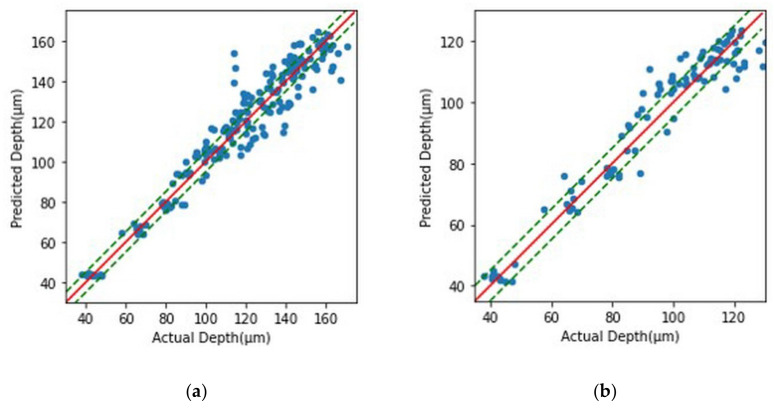
Comparison of actual data and data predicted by the learning model, and the dotted line is the point with a prediction error of ±5 μm. (**a**) Comparisons under conditions of 1 to 20 processing times. (**b**) Comparison under conditions of 1 to 10 processing times.

**Table 1 micromachines-14-00743-t001:** Number of hidden layers and the MAE value for each node condition.

Number of Hidden Layers	Number of Nodes	Training & Validation	MAE
2 hidden layers	4 nodes	Training	0.1594
		Validation	0.1896
	8 nodes	Training	0.1581
		Validation	0.1915
3 hidden layers	4 nodes	Training	0.1573
		Validation	0.1961
	8 nodes	Training	0.1543
		Validation	0.1993

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
