# Peer review of "Hole Depth Prediction in a Femtosecond Laser Drilling Process Using Deep Learning"

_micromachines, 2023, doi:10.3390/mi14040743_

Round 1

Reviewer 1 Report

In general, this is a useful addition to the literature. However, my main suggestion is to carefully discuss the industrial implications for this work. For example, figure 11 shows the main result – but how useful is this in a practical experiment. Can you apply this to a new setup, e.g. with a different camera and illumination? Or would you need to train the network again? I think some additional discussion around this area would be useful for the reader.

Introduction is strong. It includes a good number of references, although the references do not appear to be that broad, and some are quite outdated. References from two other international groups working in deep learning and femtosecond laser machining could also be added, such as some of the following recently demonstrated results.

Tani, Shuntaro, and Yohei Kobayashi. "Ultrafast laser ablation simulator using deep neural networks." Scientific Reports 12.1 (2022): 5837.

Xie, Yunhui, et al. "Deep learning for the monitoring and process control of femtosecond laser machining." Journal of Physics: Photonics 1.3 (2019): 035002.

Shimahara, Kohei, et al. "A deep learning-based predictive simulator for the optimization of ultrashort pulse laser drilling." Communications Engineering 2.1 (2023): 1.

McDonnell, Michael DT, et al. "Machine learning for multi-dimensional optimisation and predictive visualisation of laser machining." Journal of Intelligent Manufacturing 32 (2021): 1471-1483.

Mills, Benjamin, and James A. Grant‐Jacob. "Lasers that learn: The interface of laser machining and machine learning." IET Optoelectronics 15.5 (2021): 207-224.

Figure 2b,d) are quite hard to read, due to the size scales. Can these be improved? Again, the contrast on figure 3 is low and hence hard to appreciate. Can you improve the clarity on this figure?

In general, you are using the laser power (e.g. 130 mW), but it would likely be useful to state the pulse energy and/or peak power.

Many of the figure X-axes are “laser iteration”, but I think “number of laser pulses” would be clearer.

Figure 7a,b) does not make sense. The graph is showing additional pulses reduces the depth of the hole – I think this needs to be made clearer.

“These values should be considered outliers and eliminated.” – it is important to justify why it is ok to remove the outliers -after all, they are part of the distribution of data in the experimental results. Can you explain your motivation here.

Can you discuss in more detail the reasoning and methodology for the training/testing dataset split?

Author Response

The response was attached with another file.

Reviewer 2 Report

The author proposed a method to predict the drilling depth of microholes with image processing and machine learning. Here are some concerns regarding this paper.

1. In the preprocessing, the data became 10x20x3 which include processed data and label data. It's still not clear how to construct this three-dimensional data.

2. After adjustment of the brightness, exposure time and gamma value, the author believe all the samples are in the same condition. How about different field of view or magnification in imaging obtaining? Any parameters have been found to be a reference for scandalizing every photo?

3. The average error is several um. For such resolution, the author didn't provide the comparison between traditional methods.

4. Using ±5um as the margin to evaluate the performance of algorithms may not be suitable for all the depth range. A ratio such as 1/e^2 should be more reasonable.

Author Response

(The authors gave the same response as above.)

Reviewer 3 Report

1-      The authors should explain what makes this work unique and how it differs from previous research.

2-      Is not clear why the authors used 233 fs, so please clarify this issue.

3-      Given that the used laser system pulse duration could be tuned to cover the pulse durations of 100 fs to 20 ps, why did you not attempt to analyze the impact of different pulse durations on the drilled hole depth measurements?

4-      Why did you not try to evaluate the effects of varying on the measured depth of the drilled hole given that the repetition rate of the laser system utilized could be tuned to cover the single-pulse to 200 kHz?

5-      Considering that the laser system being employed has a repetition rate of 50 KHz, please detail any potential thermal effects on the depth of the drilled hole.

6-      Nothing is mentioned in the text about the uncertainty in the measurements, so please discuss this issue in the text.

7-      Please add  the amount of time that passed following "Times" in Figs. 3b and c.

8-      Please combine the two figures “Figs. 3b and c” in one.

Author Response

(The authors gave the same response as above.)

Reviewer 4 Report

1. In Introduction part, the description about the femtosecond laser is too simple. The following literatures (10.1016/j.cja.2021.01.003, 10.1016/j.actamat.2020.04.058) could give the authors more insights into this field.

2.In Experiment part, the detailed information of specimens and laser processing paraments is absent.

3. In Fig.4(a), why the depth would decreases when the laser iteration exceeds 15 time?

4. In Fig.4(b), the scale bar is absent.

5. The laser processing parameters used in Fig.5(b) should be given.

6. In Fig.7(a), why the error (exposure time :50 ms) would increase when the laser iteration exceeds 15 time?

7. what is the masking technique?

8. In Fig.11, which one is predicted value? and which one is actual value?

9. The conclusion is too long. The author should list several important points.

Author Response

(The authors gave the same response as above.)

Round 2

Reviewer 3 Report

In response to my previous suggestions and concerns, the authors have made reasonable changes to the manuscript. Overall, the manuscript reads well and clarifies the authors' work. In my opinion, the manuscript contains currently all information and is ready for publishing in the Journal " micromachines " as a regular article.

Author Response

Q1: The authors should explain what makes this work unique and how it differs from previous research.

  • There are many other papers using deep learning techniques by adjusting laser parameters and processing parameters. These papers did deep learning to derive outputs for the actual measured data values. However, the 2D image contains not only the laser parameters, but also the parameters of the measurements, such as the brightness of the illumination that occurred when the image was acquired. In this paper, the similarity of curves is derived and predicted between the actual depth curve and grayscale values of the number of processing cycles. I think that is our strength.

Q2: Is not clear why the authors used 233 fs, so please clarify this issue.

  • In micro laser machining application, the femtosecond laser is now widely used. The high peak power can apply to the certain region and it can minimize thermal effect to surrounding region. Recently, hole drilling in probe card application, the fs laser is quite well used.

Q3: Given that the used laser system pulse duration could be tuned to cover the pulse durations of 100 fs to 20 ps, why did you not attempt to analyze the impact of different pulse durations on the drilled hole depth measurements?

  • The laser model is PH2-10W, and the pulse duration can be varied from 290 fs to 10 ps based on the specification data sheet. Currently, our laser uses a pulse duration of 233 fs.
  • We plan to experiment with the measurement results of laser drilling depth by pulse duration in the future. Thank you for your good advice.

Q4: Why did you not try to evaluate the effects of varying on the measured depth of the drilled hole given that the repetition rate of the laser system utilized could be tuned to cover the single-pulse to 200 kHz?

  • The laser equipment we use is capable of variable repetition rates from 10kHz to 600kHz. However, when the repetition rate increases on the 50 kHz basis, the pulse energy decreases, and when the repetition rate decreases, only the repetition rate changes while the pulse energy remains the same. In addition, when the pulse energy becomes low, a phenomenon occurs in which the inside of the hole changes as shown in Fig. 4(b). This phenomenon is now regarded as a melting phenomenon, and a detailed analysis will be conducted in the future. And in order to secure more data in the future, we will proceed with change experiments when performing laser processing holes at 50 kHz or less. Thank you for your good advice.

Q5: Considering that the laser system being employed has a repetition rate of 50 KHz, please detail any potential thermal effects on the depth of the drilled hole.

  • The femtosecond laser theoretically has a very short pulse duration and is known to have almost no thermal effect on the sample during processing. However, in actual processing, laser beams are overlapped during processing, which causes thermal effects. In this paper, we proceeded with an experiment in which the depth of the hole changed according to the number of processing cycles. As the holes depth increases, the number of overlapping laser pulses increases. Therefore, we think that the more processing cycles, the greater the potential thermal effects.

Q6: Nothing is mentioned in the text about the uncertainty in the measurements, so please discuss this issue in the text.

  • The uncertainty of hole depth measurement was depicted in Figure 7(a)

Q7: Please add  the amount of time that passed following "Times" in Figs. 3b and c.

  • Figure (b) and (c) have been combined and the processing part has been modified to make it easier to see. And pulse energy was also added in Page 5.

Q8: Please combine the two figures “Figs. 3b and c” in one.

  • Figure (b) and (c) have been combined and the processing part has been modified to make it easier to see. And pulse energy was also added in Page 5.

Reviewer 4 Report

1. In intrduction part, the background of the femtosecond laser is still not comprehensive. please read the literatures provided in the first round of review carefully. The number of the referenced literatures is only 18. And some important literatures about femtosecond laser techniques are not cited.  It is not appropriate. 

2. Some explanations for the reviewer's comments has not been added into the revised manuscript.

Author Response

Q1: In Introduction part, the description about the femtosecond laser is too simple. The

following literatures (10.1016/j.cja.2021.01.003, 10.1016/j.actamat.2020.04.058) could give the

authors more insights into this field.

  • Corrected additional description of femtosecond laser on page 1 last line and page 2 first line.

Q2: In Experiment part, the detailed information of specimens and laser processing paraments

is absent.

  • Added information about specimens and information about laser parameters. Page 3.

Q3: In Fig.4(a), why the depth would decrease when the laser iteration exceeds 15 time?

  • Actually the hole depth with number of process cycles is nonlinear. Because it can be measured that the depth decreases as the hole fills with molten material as the number of cuts increases when the laser pulse energy is low.

Q4: In Fig.4(b), the scale bar is absent.

  • Added the scale bar in figure 4(b). Page 6.

Q5: The laser processing parameters used in Fig.5(b) should be given.

  • Added laser processing parameters in figure 5. Page 6.

Q6: In Fig.7(a), why the error (exposure time :50 ms) would increase when the laser iteration

exceeds 15 time?

  • Instead of the error bars increasing at 50ms, the actual measurement depth error bars appear to overlap. The reason why the error bar increases is that the sample was measured with the melting phenomenon inside the hole due to the laser power, so the error bar increases. I also changed the colors of the graph data.

Q7: what is the masking technique?

  • A rectangular area with 70% of a hole area was extracted relative to the center of the hole in order to obtain only the data inside the hole. When data was obtained in the diameter direction of the hole, the contrast data was not uniform at the edge of the hole due to burrs, so a rectangular area was set that did not include the edge.
  • The rest of the data except the set area was deleted and the contrast value was obtained by averaging the data of the set area.

Q: In Fig.11, which one is predicted value? and which one is actual value?

  • X axis mean actual measurement value, y axis mean data predicted value using deep learning model.
  • The red line is the line where the predicted data matches the actual data.
  • If the actual measured value at any one point is 100μm, we can think that the closer the red line is between the many y-value lines at x = 100, the more accurate the prediction.

Q9: The conclusion is too long. The author should list several important points.

  • It was revised as your suggestion.